# Current Technologies for Managing Type 1 Diabetes Mellitus and Their Impact on Quality of Life—A Narrative Review

**DOI:** 10.3390/life13081663

**Published:** 2023-07-30

**Authors:** Viviana Elian, Violeta Popovici, Emma-Adriana Ozon, Adina Magdalena Musuc, Ancuța Cătălina Fița, Emilia Rusu, Gabriela Radulian, Dumitru Lupuliasa

**Affiliations:** 1Department of Diabetes, Nutrition and Metabolic Diseases, “Carol Davila” University of Medicine and Pharmacy, 8 Eroii Sanitari Blvd., 050471 Bucharest, Romania; viviana.elian@umfcd.ro (V.E.); emilia.rusu@umfcd.ro (E.R.); gabriela.radulian@umfcd.ro (G.R.); 2Department of Diabetes, Nutrition and Metabolic Diseases, “Prof. Dr. N. C. Paulescu” National Institute of Diabetes, Nutrition and Metabolic Diseases, 030167 Bucharest, Romania; 3Department of Microbiology and Immunology, Faculty of Dental Medicine, Ovidius University of Constanta, 7 Ilarie Voronca Street, 900684 Constanta, Romania; 4Department of Pharmaceutical Technology and Biopharmacy, Faculty of Pharmacy, “Carol Davila” University of Medicine and Pharmacy, 6 Traian Vuia Street, 020945 Bucharest, Romania; catalina.fita@umfcd.ro (A.C.F.); dumitru.lupuliasa@umfcd.ro (D.L.); 5Romanian Academy, “Ilie Murgulescu” Institute of Physical Chemistry, 202 Spl. Independentei, 060021 Bucharest, Romania; amusuc@icf.ro; 6Department of Diabetes, N. Malaxa Clinical Hospital, 12 Vergului Street, 022441 Bucharest, Romania

**Keywords:** type 1 diabetes mellitus, insulin, diabetes management, technology, continuous glucose monitoring systems, insulin pumps

## Abstract

Type 1 diabetes mellitus is a chronic autoimmune disease that affects millions of people and generates high healthcare costs due to frequent complications when inappropriately managed. Our paper aimed to review the latest technologies used in T1DM management for better glycemic control and their impact on daily life for people with diabetes. Continuous glucose monitoring systems provide a better understanding of daily glycemic variations for children and adults and can be easily used. These systems diminish diabetes distress and improve diabetes control by decreasing hypoglycemia. Continuous subcutaneous insulin infusions have proven their benefits in selected patients. There is a tendency to use more complex systems, such as hybrid closed-loop systems that can modulate insulin infusion based on glycemic readings and artificial intelligence-based algorithms. It can help people manage the burdens associated with T1DM management, such as fear of hypoglycemia, exercising, and long-term complications. The future is promising and aims to develop more complex ways of automated control of glycemic levels to diminish the distress of individuals living with diabetes.

## 1. Introduction

According to the World Health Organization, diabetes is a chronic, metabolic disease characterized by elevated levels of blood glucose, which leads over time to severe damage to the heart, blood vessels, eyes, kidneys, and nerves [1]. With more than 420 million people living with this disease worldwide, diabetes is at the top of global death causes [2].

Type 1 diabetes mellitus (T1DM) occurs when insulin production is severely impaired by the pancreatic beta cells’ autoimmune destruction [3]. Patients are usually diagnosed when 80–90% of the beta cells are destroyed. Recently, Roep et al. [4] proposed an alternative point of view in which abnormal β-cells are the main contributors to T1DM pathogenesis. Based on previously reported results of immunotherapy in T1DM [5,6,7], they showed that β-cells with limited self-defense are vulnerable to biosynthetic stress [4].

Novel strategies for preventing and managing T1DM are investigated [8,9]. One research direction in this field is antigen vaccination (with oral insulin or peptides) and stem-cell-based replacement therapies [10] to prevent and reverse diabetes autoimmunity. FDA approved in November 2022 the first drug—Tzield (teplizumab-mzwv) injection—that can delay the onset of T1DM [11,12,13], resulting in years of disease remission [14]. Data also suggest that verapamil and glucagon-like peptide-1 receptor agonists could act as beta cell protective agents (in stress conditions).

Even though there are new therapeutical approaches and promising results, insulin remains the only long-term effective treatment for T1DM. Commonly, insulin is administered as multiple daily subcutaneous injections [15,16] with doses adjusted upon the determined glycemic level (fasting, preprandial, or at the time of the injection) and the meal carb count. It is also possible to deliver insulin by continuous subcutaneous insulin infusion [17] or insulin pump.

There are concerns related to access to insulin for people living with T1DM [18] in countries with limited government expenditures for health because they must pay out-of-pocket for all or part of their diabetes care, including insulin and syringes, blood glucose meters, delivery devices, and necessary health education [19,20]. Some patients reported on specific questionnaires cost-related underuse of insulin or cost-related change in insulin type, a scenario caused by low personal income. More than a third of them did not discuss this problem with their physicians. Data suggested that these individuals have poor glycemic control and are exposed to complications [21].

T1DM remains a chronic disease associated with high morbidity [22], mortality, and a diminished work life [23,24,25]. When diabetes coexists with other diseases [26,27,28,29,30,31], the management of a patient’s condition is even more difficult.

The previously mentioned aspects have complex implications on psychology and quality of life (QoL) [32]. The risk of developing psychiatric comorbidities (depressive disorders) increases with the earlier onset of T1DM [33,34]. These conditions are defined as diabetes distress (DD) and are linked to poor compliance, low glycemic control, and long-term complications (Figure 1) [33,35].

Generally, younger people [36,37], women [38], and people with poor glycemic control [39] have an increased risk for high levels of DD. Older people have a lower DD incidence [40].

T1DM management includes complex and precise self-care measures during a person’s entire life. This continuous process could overwhelm them, making them angry, anxious, frustrated, and/or discouraged [41].

Due to the previously described conditions, the new technologies for DM management must provide clinical advantages and high usability without compromising safety [42]. Adherence and compliance to the prescribed therapy are also essential, thus improving their QoL [43]. Moreover, the accessibility of the new system through healthcare coverage [44] is critical; they must be able to afford the new devices and accessories and their maintenance [45,46].

In this context, our work aims to analyze the most recent technologies regarding continuous glucose monitors (CGMs) and continuous subcutaneous insulin infusion (CSII) systems, their psychosocial impact [47], and their capacity to improve T1DM patients’ life quality [48] in a narrative review.

We performed a literature review searching original published articles on type 1 diabetes in humans. We used a data filter on insulin therapy, insulin pumps, and advanced technologies in medical devices for insulin infusion. The selection targeted the publications in scholarly peer-reviewed journals in the past 10 years (2013–2022) written in English (no country restriction). The initial selection was from Google Scholar search. Due to the high document variety, we accessed three other primary databases: PubMed^®^/MEDLINE, ScienceDirect, and Web of Science^®^. The following keywords were used: type 1 diabetes mellitus technology (3046 journal articles), continuous glucose monitoring system (4273), continuous insulin infusion (7150), automated insulin delivery (533), patch insulin pump (153), closed-loop insulin pump (874), hybrid closed-loop insulin pump (218), artificial pancreas (3389), bionic pancreas (76). We realized our search only on articles edited in the English language and performed the final search on 15 April 2023.

## 2. Continuous Glucose Monitoring Systems

Monitoring glycemia rigorously is essential for T1DM management. Adjusting insulin doses requires six or more measurements of blood glucose level daily. The basic technology in determining blood glucose, still widely used around the world, is represented by the blood glucose meters (BGM) [49]. These devices are not very expensive and easy to use by individuals to perform different changes in the day-to-day management of their diabetes [50].

Daily glucose level monitoring was significantly improved through advanced technologies that offer continuous glucose monitoring (CGM) systems [51,52,53]. Two types of CGMs are available: professional CGM (data can only be downloaded and seen by the physician) and personal CGM (data seen in real time by the person wearing the device) comprising intermittently scanned CGMs (IS-CGM) [54] and real-time ones (RT-CGM) [55]. CGMs are based on minimally invasive [56] and non-invasive [57] techniques.

An IS-CGM consists of a sensor that measures glucose levels in the interstitial fluid every minute and stores data automatically every 15 min without transmitting them [58,59]. Data can be seen when the person scans the sensor with a reader or a smartphone.

RT-CGMs automatically measure glucose levels day and night and transmit them every 1/5 min to a receiver (reader, smartphone, insulin pump) [60,61]. They have alarm systems for out-of-range glucose levels and inform patients/caregivers about immediate/long-term events (hyperglycemia or hypoglycemia) [62]. Knowing glucose levels in real-time can help make more informed daily decisions about food balance, physical activity, and insulin doses [63].

### 2.1. Description

A CGM system comprises a sensor, transmitter, receiver, application software, and insertion tool. CGMs evaluate glucose levels in the interstitial fluid (ISF) from the subcutaneous adipose tissue [64]. The measured glucose concentrations in ISF are processed through various algorithms [65] to predict blood glucose levels (BGL).

A CGMS performance is evaluated by the mean absolute relative difference (MARD) [66]. The MARD calculation involves temporally matched glucose data from CGM systems compared to BG measurements of all subjects included in a clinical study. A high-performance CGM implies MARD value < 10%. Therefore, the BG evaluation 2–4 times daily with a glucometer ensures an optimal calibration of CGM systems, even for those without mandatory calibration, thus avoiding significant differences between CGM data and BG values.

The clinical utility of CGMs consists of therapeutic decisions [67] based on real-time glucose evaluation:▪Insulin bolus size calculation both in MDI and CSII regimens [68];▪Carbs intake for preventing/treating hypoglycemia;▪Insulin corrections for increased/increasing glycemia;▪Regulation of basal rates in sensor-augmented insulin pumps [69] or automated insulin delivery (AID) systems [70];▪Micro-bolusing in AID.

Different companies commercialize minimally invasive continuous glucose monitoring devices (MID): GlySens Incorporated (San Diego, CA, USA), Senseonics Holdings, Inc. (Germantown, MD, USA), Abbott Laboratories (Chicago, IL, USA), Medtronic plc (Minneapolis, MN, USA), F. Hoffmann-La Roche Ltd. (Basel, Switzerland), Dexcom, Inc. (San Diego, CA, USA), A. Menarini Diagnostics (Florence, Italy), LifeScan IP Holdings LLC (Malvern, PA, USA), Echo Therapeutics, Inc. (Philadelphia, PA, USA), Johnson & Johnson (New Brunswick, NJ, USA), Terumo Corporation (Shibuya City, Japan), and B. Braun Melsungen AG (Melsungen, Germany). In Table 1**,** we have exemplified the latest and most frequently used CGMs in the EU and the USA.

In Table 1, the most known FDA-approved CGMs are comparatively analyzed.

CGMS should not be exposed to magnetic resonance imaging (MRI) equipment, diathermy devices, or other devices that generate strong magnetic fields (for example, X-ray, CT scan, or other types of radiation). Exposure to a strong magnetic field has not been evaluated and can cause the device to malfunction, result in serious injury, or be unsafe.

Registered data show that the most complex and expensive CGM is Eversense E3 CGM. It also has the most prolonged period of sensor-wearing but is suitable only for adults. The other CGMS have RT under 15 days but can also be used for children.

The measured glucose levels are strongly influenced by numerous factors related to the insertion place [71]. The abdominal region [71] is the most common location, followed by the upper arm. Measurements from the right abdominal site tend to be more diminished than those from the left. Studies on CGM sensors involving sleep [72] and belt compression [73] reported a potential influence of diminished blood flow on the sensors’ measurements. Similarly, Mensh et al. [64] demonstrated that the susceptibility of CGMs to abnormal nocturnal glucose readings was related to sleeping positions. On the abdominal surface, various physiologic conditions [74,75] could be related to different CGM readings in the left versus right part. In various BMI values, the subcutaneous adipose tissue size could influence CGM measurement accuracy [76].

The sensors’ biocompatibility could determine the discrepancy between abdominal sites [77]. Sensor insertion traumatizes the afferent zone, inducing inflammatory reactions [78], with increased blood flow and glucose [79]. Local proteolytic enzymes and reactive oxygen species [80] could negatively affect the sensors. During wound healing, capillary neoangiogenesis can supply additional glucose at the insertion site [25].

Recently, a non-invasive CGM manufactured by Afon Technology Ltd. (Caldicot, Monmouthshire, UK) is in the stage of clinical studies. It could be available soon as an alternative technology to traditional blood glucose monitoring tools. This CGM system utilizes high-frequency microwaves, a technology not yet used in this application [81].

The most important features regarding used materials, techniques, and methods are displayed in Figure 2.

### 2.2. Benefits

Compared with a standard blood glucometer, every day using a CGM system can help to:▪Maintain a constant glycemic level daily;▪Diminish time spent in hypoglycemia and severe hyperglycemia;▪Reduce the number of finger pricks;▪Decrease BG variability and HbA1c levels.

### 2.3. Limits

▪CGM systems are more expensive than standard glucometers;▪The finger prick glucose test is needed twice daily for some CGMs to calibrate;▪Sporadic, unpredictable errors [66];▪Invasiveness;▪Short lifespan;▪Biocompatibility.

### 2.4. Potential Adverse Effects Related to the Insertion, Removal, and Wear of the Sensor

▪Allergies to adhesives;▪Bleeding and bruising;▪Infection, pain, or discomfort;▪Sensor destruction during extraction;▪Skin inflammation, scarring, thinning, discoloration, or redness.

Other potential unwanted effects are associated with decisions made in the case of inaccurate device measurements:▪Excessive insulin administration could increase the risk of hypoglycemia;▪Inappropriate administration of carbohydrates increases the risk of hyperglycemia and acute diabetic ketoacidosis;▪Inaccurate calculation of the glucose change rate could increase the incidence of hypo or hyperglycemia.

## 3. Continuous Subcutaneous Insulin Delivery Systems (CSII)

An insulin pump is an electronic device that releases rapid insulin according to the body’s daily needs [82,83].

Insulin pumps deliver insulin in two primary ways:▪A continuous infusion of small amounts of rapid insulin throughout the day and night (basal rate);▪A one-time dose of rapid-acting insulin for meals or high blood glucose correction (bolus).

The ideal individuals for insulin pump use are:People with T1DM or insulin-dependent T2DM;People with multiple-day injections of insulin;People who can assess appropriate blood glucose control;Capable of performing insulin pump therapy initiation and maintenance;Able to maintain frequent contact with the healthcare team;Able to consider insulin pumps as a tool to improve diabetes care;Capable of accurately calculating carbohydrates and insulin bolus;Individuals with critical clinical conditions who have serious difficulties controlling glycemic targets, despite intensive treatment and monitoring;With substantially decompensated diabetes (frequent severe hypoglycemia and/or hyperglycemia);Other associated conditions: extreme insulin sensitivity, gastroparesis, pregnancy, variable schedules or work shifts, significant “dawn phenomenon”, high insulin dose therapy, or severe insulin resistance.

### 3.1. Conventional Insulin Pumps

An insulin pump is a small digital device that ensures a continuous infusion of rapid-acting insulin (CSII). The infusion set is inserted into the subcutaneous tissue and fixed on the skin with an adhesive. In most insulin pumps, the infusion set connects to the pump by plastic tubing. Insulin infuses from the pump through the tubing to the infusion set cannula and into the subcutaneous tissue. The most common devices are displayed in Table 2.

Table 3 shows the benefits and limitations of CSII pumps [84].

### 3.2. Insulin Patch Pumps (PPs)

Some insulin pumps are directly attached to the skin (patch insulin pumps). A hand device controls insulin delivery in a PP; however, some devices also allow at least some functionality via the PP. The simple forms of PPs intended for insulin therapy aim to be small and disposable, and easy to handle and carry. There are three categories of PPs: PPs with reduced features, fully equipped PPs, and PPs suitable for automatic insulin delivery (AID) systems.

The reduced-features PP delivers only basal insulin. A fully equipped PP delivers a variable amount of basal insulin over 24 h and has a bolus button that permits prandial insulin to be given in two-unit increments daily (Figure 3 and Appendix A).

Full-featured electromechanical patch pumps generally have an electromechanical structure with an electronic controller. These are all full-featured pumps with different basal rates, individually controllable bolus amounts, and additional means of bolus delivery.

PPs are small, easy to use, and discreet to wear. Moreover, they can interact with the CGM and AID systems’ algorithms. Their benefits and limitations are displayed in Table 4.

### 3.3. Sensor-Augmented Pump (SAP)

An SAP is a CSII that can integrate data from a CGM system. Glycemia is displayed on the pump in real time. It is used by the pump algorithm to automatically stop the basal insulin infusion (for up to 2 h) as a response to detected/predicted hypoglycemia. Then, the basal insulin infusion is released at the previously programmed rate. This feature helps diminish moderate-to-severe hypoglycemia, especially during nighttime, and reach better glycemic control [88]. SAPs are known as open-loop systems [89].

There are two types of SAPT available on the market:▪Low-glucose suspend (SAPT-LGS): Suspends basal rate when hypoglycemia occurs.▪Predictive low-glucose management (SAPT-PLGM): Can suspend basal rate before hypoglycemia occurs.

SAP can reduce hypoglycemia by 40–50% (<70 mg/dL) without an increase in glycosylated hemoglobin [90,91,92,93].

### 3.4. Closed-Loop Insulin Systems (Artificial Pancreas)

A CGM could become a part of a CSII through an algorithm, generating a closed-loop insulin system (Figure 4A). It is a substantially improved system, adjusting insulin delivery in response to real-time sensor glucose levels and other inputs, such as meal intake (Figure 4B). Control algorithms can modulate the insulin needs’ variability between and within individual users, considering CGM accuracy limitations and insulin delivery imprecision [94].

The three main components of a closed-loop system are (Figure 4A):▪Glucose measuring device (CGM);▪Control device for BG analysis and insulin dosing regulation (computer/microprocessor);▪Insulin infusion device (insulin pump).

The control algorithms are continuously adapted to physiological changes with real-time adjustment of closed-loop control parameters (Figure 5).

Various control algorithms were developed [97]: model predictive iterative learning control (MPC) [98,99,100], proportional integral derivative (PID) controllers [101,102], and fuzzy logic control approaches [103,104].

The closed-loop system functions as a pancreas that controls BG levels. Thus, the closed-loop insulin system is known as the artificial pancreas (AP) [105]. When an AP system requires counting and registering the carbohydrate amount from mealtime, it is called a “hybrid” [106] because a part of insulin is provided automatically, and another is infused based on the reported information.

In 2016, the FDA approved the first hybrid closed-loop system [107]. It automatically gives a suitable amount of short-acting insulin at a basal rate. The patient still needs a glucose meter a few times daily, manually adjusting the insulin delivery at mealtimes and when it requires a dose correction. Nowadays, there are several FDA- and EMA-approved systems: Medtronic 770/780G, Tandem Control IQ, Omnipod 5, CamAPS, Diabeloop, etc. (Table 5).

Another artificial pancreas system is known as a hormonal bionic pancreas (BP) [109,110,111]. It has the next-generation technology to deliver insulin and/or glucagon automatically rather than standard-of-care management. Therefore, BP is more effective in maintaining blood glucose levels within the normal range in T1DM people [112]. In May 2023, the Beta Bionics Inc. (Boston, MA, USA) iLet ACE Pump and iLet Dosing Decision Software pancreas system [109,111,113,114,115] received FDA approval.

Like a healthy pancreas, an utterly automated closed-loop system does not request meal announcements; it can react to BG level variations [116]. The benefits and limitations of closed-loop systems are given in Table 6.

#### Complications

▪Hypoglycemia occurs when a significant basal rate of insulin is delivered due to a human error in insulin pump programming or a device malfunction.▪Hyperglycemia is caused by programming error or device malfunction, leading to a low insulin delivery rate (battery depletion or malposition, cannula occlusion, total pump failure).▪If the infusion set is not changed regularly, at 3–4 days, there are irritation and infections at the place of cannula insertion.▪Insulin pump therapy discontinuation (18–50%) is the T1DM patient choice for various reasons: unwanted interference with the lifestyle, missing improvements in glycemic control, and infection at the insertion place. It occurs with high incidence in women, younger individuals, pregnancy, and when the patient has psychological comorbidities.

## 4. Decision Support Systems

As T1DM individuals play a substantial role in their healthcare, artificial intelligence applications known as decision support systems (DSSs) can help them in decision making and reacting in real time to maintain normal BGLs [117]. The DSSs notably impact the users’ decision making, increasing their confidence in their potential to deal with T1DM (Figure 6). They integrate data such as insulin-to-carb ratio, insulin sensitivity, insulin time of action, sick days, and exercise response. A DSS suggests the best therapeutic option for better glycemic control, higher time in range, diminished hypoglycemic events, and decreased HbA1c.

## 5. The Impact of New Technologies on T1DM People’s Quality of Life

### 5.1. Evaluation of Diabetes Distress

The T1DM adult questionnaire has a self-report scale with 28 items, evidencing seven critical points of diabetes distress. These seven items refer to crucial feelings of T1DM adults, according to Fisher et al. [119] (Figure 7).

Open discussions with T1DM subjects represent a clinical instrument for their emotional behavior evaluation.

### 5.2. Satisfaction Survey for Diabetes Technology Users

The Glucose Monitors Satisfaction Survey (GMSS) [120] was conceived for T1DM individuals as a 15-item self-report scale, with three key dimensions:▪Openness;▪Emotional and behavioral burdens;▪Trust.

The Insulin Device Satisfaction Survey (IDSS) is a self-report scale with a 14-item version for T1DM [121], with three key dimensions:▪Effectiveness;▪Burdensomeness;▪Inconvenience.

All these documents are available in various languages and accessible to non-profit institutions for use in clinical care and research. Concomitantly, they can be accessed as copyright with licensing fees by all for-profit companies and institutions on https://behavioraldiabetes.org/scales-and-measures/ (accessed on 9 April 2023).

### 5.3. Quality of Life Evaluation

Joensen et al. [122] analyzed the effect of continuous care and support for T1DM adults with insulin pump therapy. The flexible and participatory peer support approach augmented individuals’ motivation and empowerment, induced a feeling of serenity about diabetes management, and demonstrated the potential to diminish diabetes distress, thus increasing their QoL. Adequate diabetes-specific social support avoids the risk of participants feeling isolated with peer support groups and allows free interactive dialogue regarding T1DM disease and all the difficulties of coping with it. Evaluation of motivation, serenity in life with diabetes, awareness of own diabetes practices, diabetes empowerment, diabetes loneliness, and diabetes distress are applicable, feasible, and appropriate when measuring the effect of peer support on adults with insulin-pump-treated diabetes.

Rizvi et al. evaluated the impact of frequent interactions between healthcare providers and individuals using CGMs in a diabetes center. It involved virtual video visits with high frequency and an electronic messaging system. CGM data were downloaded proactively every 2 weeks from computer cloud-based device accounts of 146 individuals with diabetes on multiple (2–5) daily insulin injections or insulin pump therapy. Then, the diabetes specialist analyzed and interpreted the glucose profile and individually advised each patient regarding diet, lifestyle, and insulin regimen. The communication used an electronic portal linked to email during telehealth and in-person office visits. As a result, the adherence was approximately 100% [123].

Introducing technology in T1DM management in individuals with a diabetes duration of over 30 years also encounters multiple obstacles. Our team’s experience materialized in a model of the efficiency of this process. Following this model and after the treatment change, we obtained a 1.3% reduction in HbA1c and significantly fewer hypoglycemic events, RR 34% (*p* < 0.01). People’s acceptance of the new treatment increased in time by 92%. The implementation process effectively achieved this transformation, reached glycemic targets, and changed patients’ lifestyles [124].

Recently, Fanzolla et al. supervised a two-section questionnaire (children/parents) to evaluate the impact of CGM on T1DM children and their families. The data show that CGM devices significantly improved the QoL of children and their parents, as 80% of children reported that CGM subcutaneous placement is much less painful than fingertips. Moreover, the QoL at school and sports is significantly better: diminished anxiety, higher comfort, and better glycemic control. Also, 90% of parents stated that CGM devices remarkably improved the glycemic control of their children, reducing emerging events. A similar percentage (89% of parents) are convinced that the recent technology generated substantial benefits for their children’s QoL [125].

Rusak et al. [126] evaluated the quality of life and satisfaction in children under 7 years of age with T1DM using the RT-CGM system integrated with an insulin pump, investigating their caregivers’ opinions. They reported high satisfaction with CGM use (68% in groups aged 5–7 years and 92% in 2–4 years), and 71% of caregivers confirmed the positive effect of CGM on their sleep quality.

Polonsky et al. [127] analyzed the CGM impact on the quality of life in T1DM adults. The participants completed a QoL evaluation (regarding overall well-being, health status, DD, confidence, and hypoglycemic fear) and a CGM satisfaction survey. CGM satisfaction was not considerably associated with glycemic levels. The satisfaction was directly correlated with diminishing diabetes distress and hypoglycemic fear and increasing well-being and hypoglycemic confidence [127]. Therefore, the CGM group reported higher hypoglycemic confidence and diabetes distress reduction than the SMBG, but with no statistically significant differences.

Fear of hypoglycemic episodes is also why people with T1DM restrain from regular physical activity [128]. However, most consider that physical exercise generates increased well-being and QOL, but there are still many barriers to engaging in physical activity. Lack of time and support from family or friends was also reported. Practicing exercise improved TIR and decreased TAR; eating before and turning off the pump during the exercise were associated with lower TBR after exercise [129].

CGM usage has proven benefits in glycemic control and decreased the frequency of hypoglycemic events during and after exercise. Pump therapy provides the tools for better insulin management by adjusting basal rates. Hybrid closed-loop systems may be the solution, but for the moment, physical exercise remains a challenge for these systems [130].

The Satisfaction Survey on Individuals With Artificial Pancreas [116] did not yet clearly reveal a clear impact on fear of hypoglycemia, adherence, quality of life, depression and anxiety, and diabetes distress.

However, T1DM children from a summer camp [131] revealed the benefits of a bionic pancreas in the following ways:▪Reducing their fear of hypoglycemia;▪Decreasing their sense of regimen burden;▪Diminishing their worries about out-of-range blood sugar levels;▪Improving their overall freedom to engage in activities that they enjoyed.

In addition, their concerns about the BP included wishing the system responded to out-of-range blood sugar levels more quickly and the annoyance of carrying several devices around [131].

## 6. Discussion

BG self-monitoring is crucial to the daily management of T1DM. However, the adherence rate is low. A recent study [132] reported a 61.6% adherence rate to the Spanish Diabetes Society protocol for SMBG. The authors identified the associated factors: the frequency of insulin injections (<3 injections daily), alcohol abstinence, peripheral vascular disease, and retrieval of the reactive strips from the pharmacy. Only 21.4% of individuals had an excellent self-perception of glycemia.

Peralta et al. [133] performed a cross-sectional observational study in adults and children treated with basal-bolus therapy and CSII users, aiming at T1DM management. They observed that metabolic control (expressed as HbA1c) positively correlates with higher educational levels, carbohydrate counting, more daily SMBG, and fewer hypoglycemic episodes. It decreases with T1DM duration, higher insulin total dose, low adherence to diet, and a family history of DM [133].

Spaan et al. [134] reported that adherence to insulin pumps in T1DM adolescents declined with age. The transition from parental care to adolescent self-caring patients led to this. Moreover, more conditions are requested to maintain the T1DM pediatric patients’ adherence to CSII therapy. Giany et al. [135] observed that insulin pumps are underused in T1DM youths. Their study integrated the evaluation of biomedical and psychosocial factors associated with consistent and durable CGM use over time. Their results were confirmed by Trandafir et al. [136], who showed that close family members (parents) and healthcare professionals have an essential role in their adherence to insulin pump therapy, together with other determinants (Figure 8).

The use of CGM by individuals with diabetes is correlated with psychological benefits and burdens [137]. Potential benefits consist of improved QoL and diminished hypoglycemia fear. However, randomized clinical trials versus cohort studies for youth and adults show heterogeneous results. In correspondent studies, adolescents and adults have nearly universally reported substantially perceived satisfaction regarding CGM use. The specific benefits were linked to easier diabetes management and better glycemic control. In contrast, limitations included pain and body issues, communication problems, feeling awestruck by the complex glucose management, and doubts regarding the accuracy of CGM compared to glucometer readings [138]. Increased anxiety of adolescents and parents and poorer parental sleep has also been remarked [139].

De Bock et al. [140] investigated users of sensor-augmented pumps in a 6-month clinical trial. Variable parameters were examined (individuals’ age and gender, the values of HbA1c, diabetes duration, frequency of BG testing, sensor accuracy, and the frequency of insulin pump alarm) and associated with CGM adherence. CGM adherence was 75% (35% to 96%), and age was the only variable significantly correlated with CGM adherence.

On the other hand, younger clinicians managing individuals who used insulin pumps and CGMs recorded more positive attitudes concerning diabetes technology [141] and identified different barriers to patient adherence: device on the body (73% pump; 63% CGM), alarms (61% CGM), and misunderstanding of the procedure (40% pump; 46% CGM). Numerous recent studies showed the positive impact of DSS and closed-loop systems on T1DM users. Thus, Breton et al. [142] reported the benefits of a DSS formed by two real-time advisors (CGM-informed bolus advisor and exercise advisor) and a retrospective insulin titration tool significantly reducing BGL variation in 48 h. In another study [117], the T1DM participants modified their initial decision 20% of the time based on DSS clinical indications. With automated exercise detection as an additional signal, a dual-hormone AP diminished hypoglycemia during exercise in the study period [143]. Technology such as CGM and pumps provides detailed information on glucose variation before, during, and after exercise and can prevent exercise-related hypoglycemic events. Also, the development of new smartphone apps designed for people with type 1 diabetes can improve time in range and diminish time below range by suggesting the appropriate amount of carbs before or after exercise and by helping the patient to adjust his/her insulin regimen [144,145]. Further planned applications of DSS will be in exercise decision and exercise as an adjunct therapy, optimizing meal bolus timing and other time-varying dosing parameters, and pregnancy. In addition, the DSS will be integrated with AID [118].

However, the healthcare resource waste associated with nonadherence to CGM and/or pump therapy and early discontinuation of device usage justifies rigorous consideration in selecting suitable patients for these technologies.

To diminish the unwanted complications of insulin administration and increase the patient’s compliance, various research teams tried to incorporate it in multiple carriers for oral administration [146,147,148]: liposomes [149,150] mixed micelles [151,152], nanoparticles [153,154,155], intestinal patches [156]. Current studies also explore other access ways for insulin administration: intranasal [157,158,159], transdermal [160,161], and sublingual [162,163].

Recent studies analyzed glucose-responsive biomaterials [163], especially for rapid and extended self-regulated insulin delivery [164]. Moreover, using chitosan hydrogels integrated with glucose-responsive microspheres for insulin delivery, Yin et al. [165] developed a synthetic artificial pancreas. However, new biomaterials or a substantial variety of composite ones are necessary to build a functional bioartificial pancreas with proper mechanical strengths and biological activities [166]. These properties can be achieved using 3D bioprinting technology. An innovative process in advanced tissue engineering aims to construct clinically applicable bioartificial pancreatic islet tissue [167] with a native tissue environment. A 3D bio-printed pancreas is expected to have critical applications for future diabetes treatment.

## 7. Conclusions

All presented data evidence that current emerging technologies and control systems significantly improve T1DM self-management. Moreover, unconditional, continuous medical and social help is essential in increasing their self-confidence, motivation, and adherence to CSII, minimizing the impact of other factors (family incomes, requested education, and the ability for technology use). The quality of life of T1DM individuals could substantially increase when the performances of advanced devices and algorithms are associated with considerable support from family and healthcare providers.

However, the technological systems’ limitations and potential adverse effects and complications lead to continuous worldwide research on finding alternative approaches to T1DM therapy.

## Figures and Tables

**Figure 1 life-13-01663-f001:**
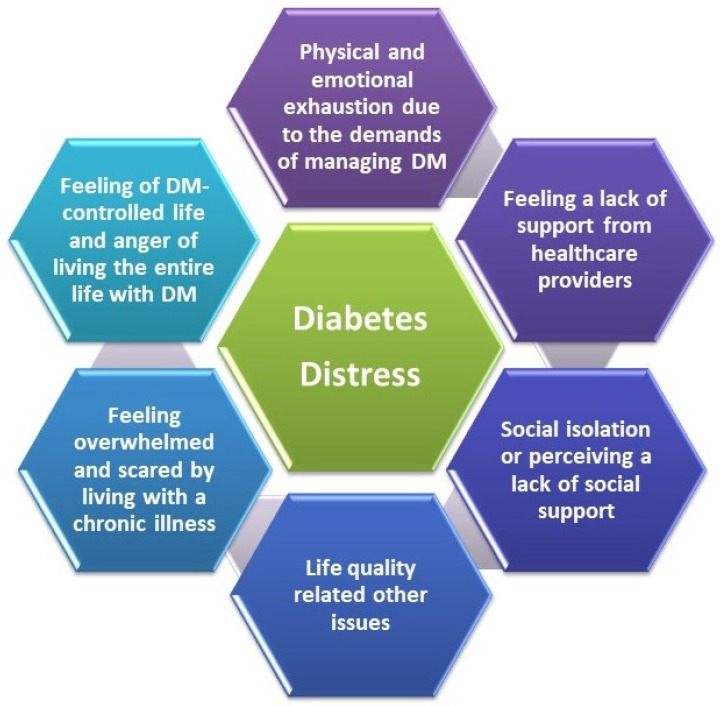
Diabetes distress, a schematic presentation; adapted from [33,35].

**Figure 2 life-13-01663-f002:**
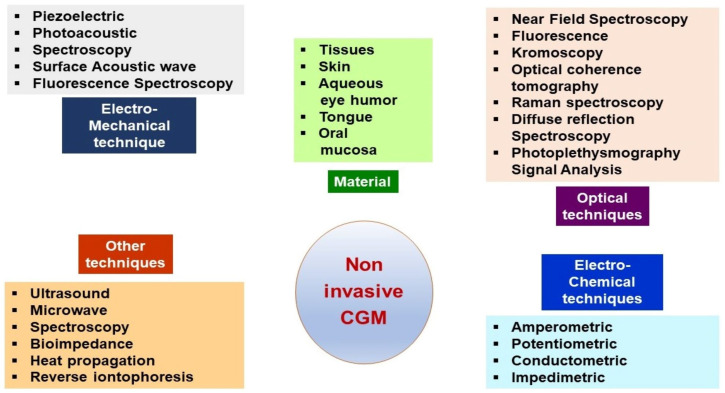
Overview of the modern non-invasive CGM approaches for intensive insulin therapy, adapted from [57].

**Figure 3 life-13-01663-f003:**
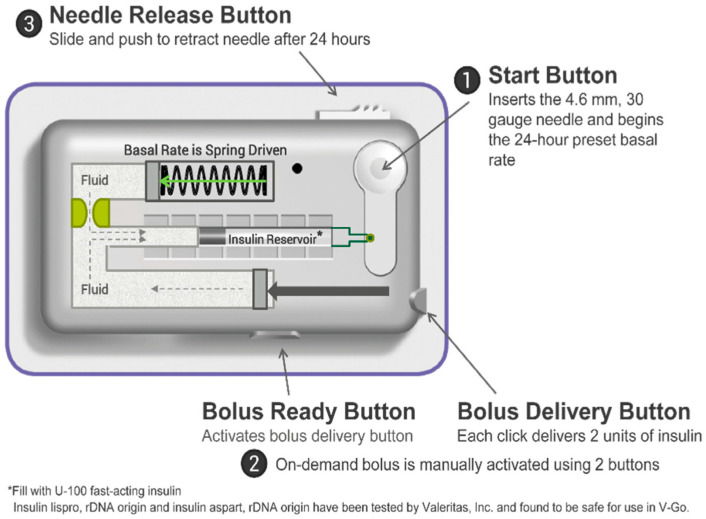
V-Go, a 24 h wearable insulin PP. Reproduction with permission from [85].

**Figure 4 life-13-01663-f004:**
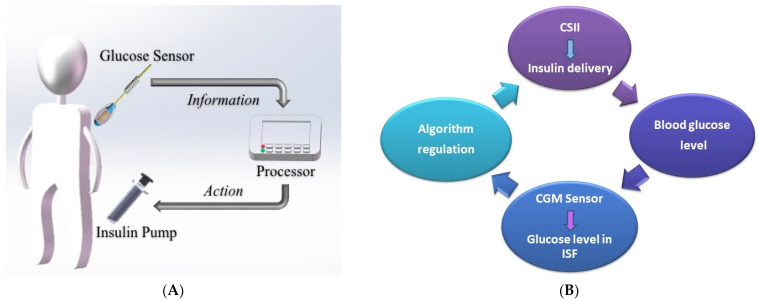
(**A**) Closed-loop glycemic management system using the “Sense and Act” method for optimal insulin delivery; reproduction with permission from [95]; (**B**) Schematic representation of closed-loop insulin delivery, adapted from [94]. A CGM wirelessly transmits the information related to interstitial glucose concentration (to a smartphone/insulin pump); the algorithm translates the received data and calculates the appropriate amount of insulin. Then, a rapid-acting insulin analog is delivered by an insulin pump. Insulin infusion is regulated in real time by the control algorithm. CSII—continuous subcutaneous insulin infusion; ISF—interstitial fluid.

**Figure 5 life-13-01663-f005:**
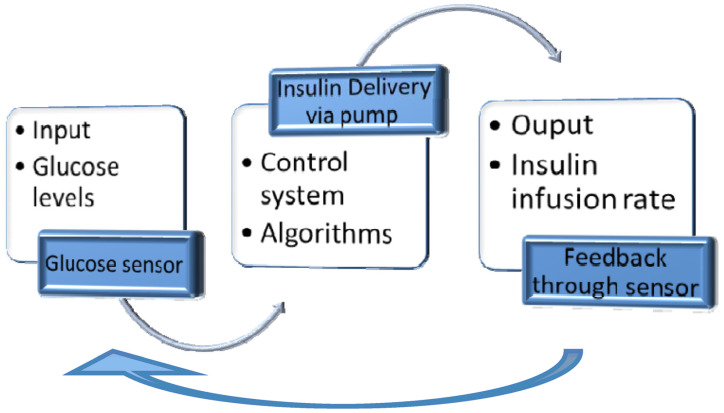
Control system principles applied to glycemic control; reproduction with permission from [96].

**Figure 6 life-13-01663-f006:**
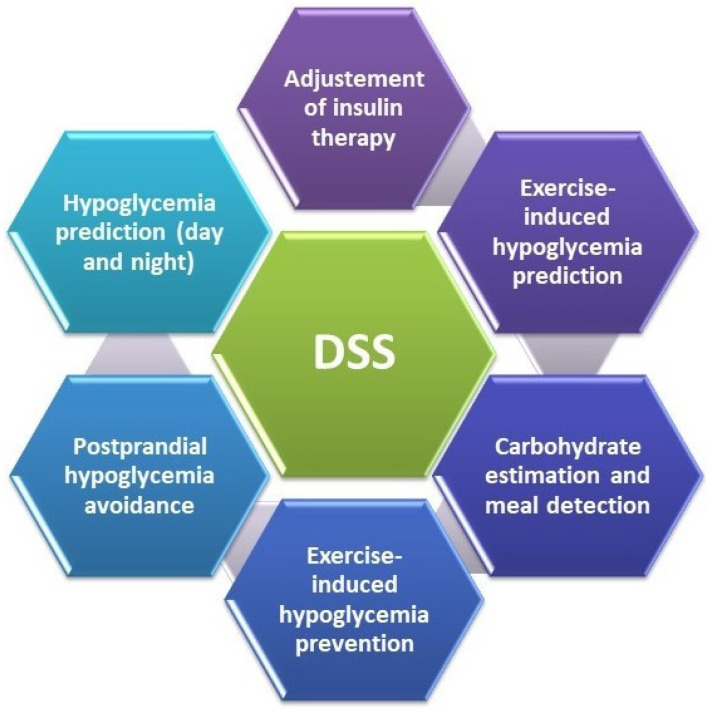
Decision support system (DSS) objectives; adapted from [118].

**Figure 7 life-13-01663-f007:**
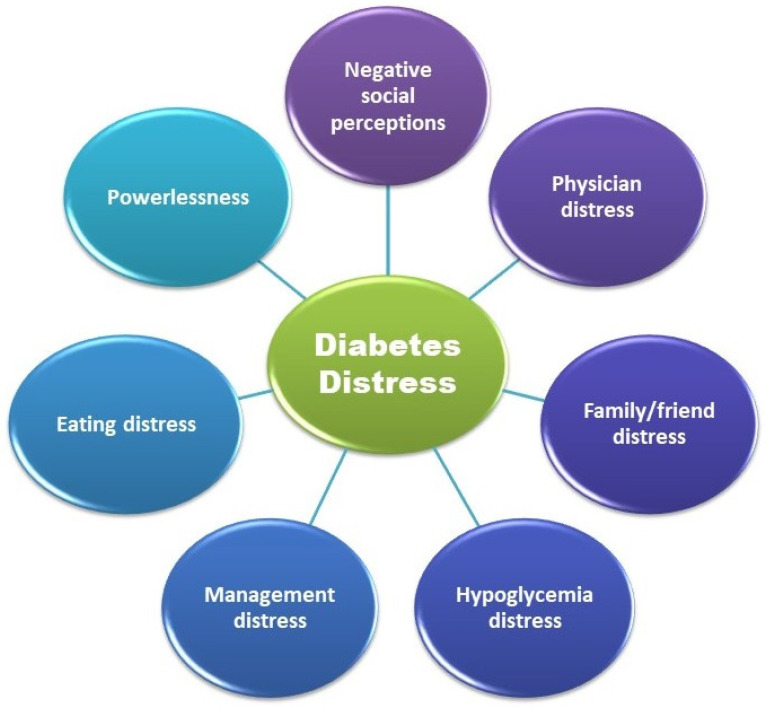
The principal sources of diabetes distress in T1DM adults, adapted from [119].

**Figure 8 life-13-01663-f008:**
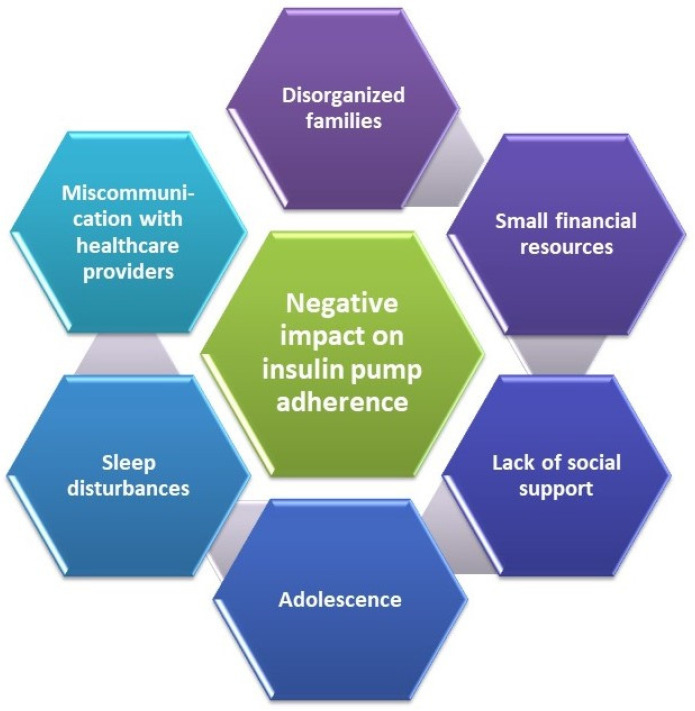
Factors negatively impacting insulin pump adherence in pediatric patients, adapted from [136]. Reproduction with permission.

**Table 1 life-13-01663-t001:** Comparison between various commercial CGM systems.

Characteristics	Dexcom G7	Guardian 4	Libre 3	Eversense E3
**Manufacturer** **[**[60]**]**	DexCom, Inc., San Diego, CA, USA	Medtronic, Minneapolis, MN, USA	Abbott Laboratories, Chicago, IL, USA	Senseonics Holdings, Germantown, MD, USA
**FDA approval**	December 2022	2021	April 2022	February 2022
**Users**	Adults and children > 2 years	Adults and children > 7 years	Adults and children ≥ 4 years Pregnant women	Adults over 18 years
**Days of sensor wear (RT)**	10 +12 h grace period	7	14	180
**Sensing molecule**	Glucose oxidase	Glucose oxidase	Glucose oxidase	Boronic-acid derivative
**Technique** **category** **[**[60]**]**	Electrochemical	Electrochemical	Electrochemical	Optique
**Components** **[**[60]**]**	Sensor, Transmitter, app	Sensor, Transmitter, app	Sensor, app	Sensor, Transmitter, app, insertion tool
**Sensor size (mm)** **[**[60]**]**	24 × 27.3 × 4.6	38 × 67 × 52 mm	21 × 2.9 × 0.11 mm	18.3 × 3.5 subcutaneous 37.6 × 48 × 8.8 transmitter
**Approved areas of insertion** **[**[60]**]**	All pts—abdomen and upper arm 2–6 years—also upper buttocks	7–17 years—upper arm, upper buttocks, Over 18 years—abdomen, upper arm	Back of upper arm	Upper arm
**Accuracy** **(MARD%)** **[**[60]**]**	8.7%	10.8%	7.8%	8.5%
**Daily calibration** **frequency (×)** **[**[60]**]**	0 (factory calibration)	0	0 (factory calibration)	2 (at 12 h)
**Warm-up** **(min)**	30	120	60	60
**High and low** **alarms**	Yes	Yes	Yes	Yes
**High and low** **prediction**	Yes	Yes	Yes	Yes
**Integration with an** **insulin pump** **[**[60]**]**	Tandem t:slim Control-IQ	Medtronic MiniMed 770G, 780G	No	No
**Smartphone** **integration**	Android, iOs, Apple Watch	Android, iOs	Android, iOs	Android, iOs, Apple Watch
**Distance to phone** **(m)**	6	6	10	-
**Operating temperature (°C)**	10–42 °C	0–45 °C	10–45 °C	5–40 °C
**Data sharing**	≤10 people	≤5 people	≤20 people	≤5 people
**A separate receiver is available**	Yes	No	Yes	No
**Water resistance** **depth (m)/time (min, hours)**	2.4 m, ≥ 24 h	2.5 m, 10 min	1 m, 30 min	1 m, 30 min
**Skin** **complications**	Yes	Yes	Yes	Yes
**Interference with drugs**	Hydroxyurea	Acetaminophen	Vitamin C	-

MARD—mean absolute relative difference; CGM—continuous glucose monitoring; data adapted from https://www.dexcom.com/en-GB/downloadsandguides/search (accessed on 7 March 2023), https://www.medtronicdiabetes.com/download-library/guardian-4-sensor-transmitter (accessed on 9 March 2023), and https://www.diabetescare.abbott/support/manuals/uk.html (accessed on 9 March 2023).

**Table 2 life-13-01663-t002:** Comparison between various commercially available CSII pumps.

	AccuChek Spirit Combo	Medtronic Paradigm 522/722	Medtronic 720G	Omipod Patch Pump	Cellnovo Insulin Pump	Dana Diabecare R
Producer	Roche Pharma	Medtronic	Medtronic	Insulet Corporation	Cellnovo	Sooil
Weight (g)	80	100	105		30	51
Dimensions (mm)	80 × 56 × 20	51 × 79 × 20	96 × 53 × 25	Pod: 41 × 61 × 18 PDA: 66 × 110 × 26	NA	54 × 75 × 19
Insulin Volume per Infusion Set (mL)	315	176-300	300	200	150	300
Basal Increments (units)	0.1	0.05	0.025	0.05	0.05	0.1
Basal Delivery minim (units)	3	10	3	3	3	3
Bolus Increments minim (units)	0.1, 0.2, 0.5, 1,2	0.1	0.025	0.05	0.05	0.1–87
Basal Rates/24 h (units)	24	48	48	48	24	48
Basal Profiles (units)	5	3	8	7	20	4
Bolus Calculator	On separate device	Yes	Yes	Yes	Yes	Yes
Multiple Bolus-type *O*ptions	Yes	Yes	Yes	Yes	Yes	Yes

NA = not available.

**Table 3 life-13-01663-t003:** Benefits and limitations of CSII pumps, adapted from [84].

Insulin Pump Therapy Benefits	Insulin Pump Therapy Limitations
**Better diabetes control.** **Fewer injections.**	The need to understand the functioning and proper management of the device.
**Improved quality of life.**	High costs, if not covered by the insurance company.
**The flexibility of basal insulin delivery during the day and night.** **The flexibility of food intake and exercises.**	A device that should be worn on the body with tubing that can be caught on objects.
**Diminished risk of hypoglycemia.**	Skin allergies or infections.
**Diminished risk of complications.**	Multiple alerts.

**Table 4 life-13-01663-t004:** Advantages and limitations of PPs, adapted from [86,87].

Advantages	Limitations
**The devices are tubeless, without request for an insulin infusion system**	Waste of insulin when PPs are replaced.
**The needle could be automatically inserted; thus, their application could be less painful.**	The infusion place is poorly visible, and regular inspection is complex.
**The needle is not visible.** **More convenient than conventional pumps for numerous activities (showering, swimming, sweating, or exercising).**	The accuracy of insulin delivery of some PPs is often lower than that of conventional pumps, particularly at low basal doses.
**Smaller and lighter than conventional pumps.** **PPs can be discreetly carried to various body parts, offering more effortless movement.** **Technical properties are often specifically adapted to T1DM individuals’ needs.** **Simple education and training are requested for their use.**	They have a poor ecological balance due to waste from plastic material and batteries. Risk of infections.
**Diminished price if certain PPs compared to conventional pumps.**	Higher price than MDI.

IIS—insulin infusion system; PP—patch pump; MDI—multiday injections.

**Table 5 life-13-01663-t005:** Comparison between various commercially available hybrid closed-loop systems [108].

	Tandem t:Slim X2 Control IQ	Medtronic 780G	Omnipod 5	CamAPS FX
**Producer**	Tandem Diabetes Care, San Diego, CA, USA	Medtronic plc, Dublin, Ireland	Insulet Corporation, Acton, MA, USA	CamDiab Ltd., London, UK
**Pump**	Tandem	Medtronic	Omnipod	Dana, Ypsopump
**CGM**	Dexcom G6	Guardian 3 and 4	Dexcom G6	Dexcom G6, Freestyle Libre 3
**CGM duration (days)**	10	7	10	10/14
**Algorithm type**	MPC	PID	MPC	MPC
**Algorithm configuration**	On pump	On pump	On pump	App on Android smartphone
**Approved for ages (years)**	6 and above	7 and above	2 and above	1 and above
**Weight (g)**	112	105	Pod 26 PDA 165	NA
**Dimensions (mm)**	79 × 51 × 15	96 × 53 × 25	Pod 39 × 52 × 14.5	NA
**Tubeless**	No	No	Yes	No
**Insulin volume per infusion set** **(mL)**	300	300	85–200	300
**Minimum basal increments (units)**	0.001	0.025	0.05	NA
**Bolus range (units)**	0–25	0–25	0.05–30	
**Minimum daily dose (units)**	10	8	6	5
**Algorithm target (mg/dL)**	110–160	100 or 120	110–150	104 (80–198)
**Meal detection**	No	Yes	No	No

Data are available in each product’s technical specifications, adapted from https://www.pcdsociety.org/resources/details/diabetes-technology-state-art, accessed on 10 March 2023. NA = not available.

**Table 6 life-13-01663-t006:** Benefits and limitations of closed-loop systems.

Closed-Loop Insulin Systems Benefits	Closed-Loop Insulin Systems Limitations
** The glucose levels can be continuously monitored. **	The T1DM patient regularly verifies the devices to ensure that they function correctly.
** The control algorithms improve BG control, automatically regulating the amount of insulin. **	The user must continuously verify the CGM and infusion pump catheter, ensuring they are in a suitable place, and change them when needed.
** The system helps the T1DM user avoid emerging events (hypoglycemia and hyperglycemia). **	The CGM accuracy should be verified, and the CGM sensor must be regularly replaced.
	The patient must count the mealtime carbohydrates and enter them into the system.
	The control software settings must be verified to ensure that the insulin infusion has a suitable amount.
	The extreme BG levels should be regulated if the system is unable.
	The pump’s algorithm could not predict exercise.

## Data Availability

Data are available in the MS and Appendix A.

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
