# Peer review of "Current Technologies for Managing Type 1 Diabetes Mellitus and Their Impact on Quality of Life—A Narrative Review"

_life, 2023, doi:10.3390/life13081663_

Round 1
Reviewer 1 Report
I want to thank for the opportunity to review this manuscript. The importance of technology support for therapy in T1DM is critical, and worth investigation, and since the new advances, it is important to provide accurate summaries of the recent findings.
However, I think that the manuscript could benefit from some suggestions that I hope can help improve its quality:
- Also, it is usually preferred to use the term "people" or "individuals" rather than "patients".
Abstract:
- I found it poorly informative. It seems it describes this work's overall aim/objectives, but they do not summarize the findings and take-home message.
- In addition, i) the authors defined T1DM but not DM; ii) "the accessibility of the new system for patients through healthcare coverage is critical" it is unclear what the authors mean - reword, please.
-Finally, the abstract should summarize the paper and it is not commonly accepted that the same sentences are used in the main text and in the abstract (e.g. "Type 1 Diabetes mellitus (T1DM) occurs when insulin is not produced in the pancreas 43 due to the beta cells' autoimmune destruction")
Introduction
- Line 81: "Some try to underuse insulin treatment without informing their clinicians about this condition [40]". Please expand, it is an important point.
- Line 84: "the patients highly depend on exogenous insulin associated with substantial lifestyle adjustments to normalize lipid and protein metabolism [44]". Again, not clear, what the authors mean by "normalize lipid and protein metabolism"?
Methods
- It is not clear if this is a narrative review or a systematic review. If the latter, the authors should provide the flowchart for inclusion/exclusion criteria, specifying how the articles were evaluated and how many authors independently assessed their quality. Otherwise, please state clearly also in the title, that this is a narrative review.
- Why only open-access articles? This is an extremely severe selection bias, as many works are not published as open-access, and many times open-access is available when fundings are present and this might affect the findings.
Discussion:
- The authors should better explain the importance of the choice of the device to promote "safe" physical activity in these people, as reduced exercise is often reported as being due to therapy management (Brazeau et al., Diabetes Care, 2008). In addition, one of the primary challenges for insulin delivery devices and glucose monitoring is related to the "exercise" condition, and to better estimate/adapt therapy in such peculiar physiological condition (Chetty et al., Front Endocrinol, 2019; Ferreira et al., Acta Diabetologica, 2023).
- Since new technologies and solutions are discussed, providing a paragraph related to the use of smart/app solutions should be recommended, as they can help people with T1DM to cope with many therapeutic challenges and they can be cheap and easily accessible for most individuals (Kordonouri and Riddell, Ther Adv Endocrinol Metab, 2019; Buoite Stella et al., Can J Diabetes, 2017).
Overall, I recommend proofreading the text since many parts could benefit from improving English grammar and spelling.
Author Response
Please, see the attachment.

Reviewer 2 Report
Dear Authors,
The paper is a narrative review of the role of technology in the management of T1D and in improving the quality of life in this clinical setting. Despite the interest, the paper should be rearranged before considering it suitable for publication.
1. Introduction: I suggest shortening it significantly, providing only basic information about the pathophysiology of T1D, the rationale for insulin replacement therapy, including strategies nowadays available for insulin delivery, and both benefits and burdens related to composite insulin therapy to briefly introduce the need for the use of technology in this setting.
2. The Materials and Methods section could be deleted, and the databases' literature searching methods could be briefly included in the introduction (after the study aim).
3. CGM section: please check for possible inaccuracies and mistakes. For example, structured measurement of glucose levels may require six or more daily checks in insulin-treated patients. Table 1 should be updated considering newer sensors, and I suggest you include a line describing the potential interaction between glucose readings and the concomitant use of certain medications (as an alternative, you could discuss it in the text).
4. Insulin pump section: a table resuming all available insulin pumps and describing essential characteristics could be helpful. Moreover, the description of devices, benefits, and limitations should be shortened.
5. Please overview data on clinical efficacy/effectiveness (based on the study design) and safety of sensors, pumps, and both (T1D patients only).
6. Discussion: make sure not to be repetitive.
Author Response
Please, see the attachment.

Round 2
Reviewer 1 Report
Thank you
I am fine with the corrections made by the authors.
Author Response
Dear Reviewer 1,
Thank you so much for all your efforts, attention, and valuable comments to increase the quality of our MS.
We are grateful for your appreciation and wish you all the best.
Reviewer 2 Report
Dear Authors,
Adequate changes have been made. I suggest revising the text for minor errors. Tables should be checked for minor changes (e.g., the unit of measurement, when necessary).
Author Response
Dear Reviewer 2,
Thank you so much for all your time, professionalism, attention, and valuable comments to increase our MS quality. Thank you again for appreciating our efforts to achieve it; we are so glad we succeeded.
All requested changes from the second review report were performed and marked in the MS text with track changes.
Moreover, we re-edited our figures in a more suitable mode with the same colors. We hope the current version is good enough to be accepted for publication in Life Journal.